# Interstitial Lung Disease at High Resolution CT after SARS-CoV-2-Related Acute Respiratory Distress Syndrome According to Pulmonary Segmental Anatomy

**DOI:** 10.3390/jcm10173985

**Published:** 2021-09-02

**Authors:** Elisa Baratella, Barbara Ruaro, Cristina Marrocchio, Natalia Starvaggi, Francesco Salton, Fabiola Giudici, Emilio Quaia, Marco Confalonieri, Maria Assunta Cova

**Affiliations:** 1Department of Radiology, Cattinara Hospital, University of Trieste, Strada di Fiume 447, 34128 Trieste, Italy; elisa.baratella@gmail.com (E.B.); m.cova@fmc.units.it (M.A.C.); 2Department of Pneumology, Cattinara Hospital, Strada di Fiume 447, 34128 Trieste, Italy; francesco.salton@gmail.com (F.S.); marco.confalonieri@asugi.sanita.fvg.it (M.C.); 3Department of Medicine, Surgery and Health Science, University of Trieste, Strada di Fiume 447, 34128 Trieste, Italy; cristinamarrocchio@gmail.com (C.M.); natalia.starvaggi@gmail.com (N.S.); 4Biostatistics Unit, Department of Medicine, Surgery and Health Sciences, Cattinara Hospital, University of Trieste, Strada di Fiume 447, 34128 Trieste, Italy; fgiudici@units.it; 5Unit of Biostatistics, Epidemiology and Public Health, Department of Cardiac, Thoracic, Vascular Sciences and Public Health, University of Padua, 35100 Padua, Italy; 6Department of Medicine-DIMED, Radiology Institute, University of Padua, Via Nicolò Giustiniani, 2, 35128 Padua, Italy; emilio.quaia@unipd.it

**Keywords:** acute respiratory distress syndrome, COVID-19 pneumonia, high resolution computed tomography, pulmonary fibrosis

## Abstract

Background: The purpose of this study was to evaluate High-Resolution CT (HRCT) findings in SARS-CoV-2-related ARDS survivors treated with prolonged low-dose methylprednisolone after hospital discharge. Methods: A total of 44 consecutive patients (M: 32, F: 12, average age: 64), hospitalised in our department from April to September 2020 for SARS-CoV-2-related ARDS, who had a postdischarge CT scan, were enrolled into this retrospective study. We reviewed the electronic medical charts to collect laboratory, clinical, and demographic data. The CT findings were evaluated and classified according to lung segmental distribution. The imaging findings were correlated with spirometry results and included ground glass opacities (GGOs), consolidations, reticulations, bronchiectasis/bronchiolectasis, linear bands, and loss of pulmonary volume. Results: Alterations in the pulmonary parenchyma were observed in 97.7% of patients at HRCT (median time lapse between ARDS diagnosis and HRCT: 2.8 months, range 0.9 to 6.7). The most common findings were linear bands (84%), followed by GGOs (75%), reticulations (34%), bronchiolectasis (32%), consolidations (30%), bronchiectasis (30%) and volume loss (25%). They had a symmetric distribution, and both lower lobes were the most affected areas. Conclusions: A reticular pattern with a posterior distribution was observed 3 months after discharge from severe COVID-19 pneumonia, and this differs from previously described postCOVID-19 fibrotic-like changes. We hypothesized that the systematic use of prolonged low-dose of corticosteroid could be the main reason of this different CT scan appearance.

## 1. Introduction

There is an association between the COVID-19 syndrome, due to the new severe acute respiratory syndrome-coronavirus-2 (SARS-CoV-2), with acute respiratory distress syndrome (ARDS), requiring noninvasive/invasive mechanical ventilation, in approximately 15% of cases [1]. ARDS is one of the most deleterious forms of acute lung injury and develops within 7 days from an identifiable cause other than cardiac failure. Initial predictive signs are bilateral radiographic lung opacities and severe hypoxemia, defined as a PaO_2_:FiO_2_ of ≤300, measured with at least 5 cmH_2_O positive end-expiratory pressure [2]. To date, the treatment for ARDS is mainly supportive and includes non-invasive mechanical ventilation (NIV), although intubation and invasive mechanical ventilation (IMV) are required in some cases [3]. After numerous randomized controlled trials demonstrated that glucocorticoids are the only drug able to improve survival in hospitalized patients, they have become the mainstay of COVID-19 treatment [4,5]. The high mortality rate of ARDS increased further during the COVID-19 pandemic to reach almost 50% of those affected. Acute phase survivors enter into an extended repair phase characterised by a decrease in alveolar infiltrates, followed by gradual clinical improvement [6]. Although quite a large proportion of these patients do return to normal lung function and imaging within the following few months, some of them have persistent interstitial infiltrates, followed by pulmonary fibrosis associated with a restrictive pattern and/or a reduction in carbon monoxide diffusing capacity (DLCO). 

Whether or not COVID-19 patients who survive the acute phase of the disease are at risk of chronic sequelae is currently a question of debate. However, 70% of patients who recover from ARDS, whatever the underlying cause, have abnormal imaging findings at a 6-month follow-up [7]. Moreover, it has been demonstrated that previous coronavirus-related epidemic infections, such as severe acute respiratory syndrome (SARS) and the Middle East respiratory syndrome (MERS), are related to a higher incidence of fibrosis [8]. Therefore, it is reasonable to presume that pulmonary fibrosis may well be a sequelae of COVID-19, and several hypotheses as to its potential pathogenetic mechanisms have been put forward [9,10].

This study describes the short-term high-resolution CT (HRCT) findings in patients with a severe COVID-19 pneumonia complicated by ARDS and treated with systemic corticosteroid according to WHO recommendations after hospital discharge. The secondary objective was to investigate the correlation between the HRCT findings and the respiratory functional tests at follow-up.

## 2. Materials and Methods

This retrospective study was approved by the Local Ethical Committee (CEUR-2020-Os-148) and carried out in accordance with the Declaration of Helsinki. A total of 44 consecutive patients, who had been admitted to the Respiratory High-Dependency Unit (RHDU) of the University Hospital of Trieste between 1 October 2020 and 30 November 2020 with a SARS-CoV-2 infection, demonstrated by a positive reverse transcriptase polymerase chain reaction (RT-PCR) test on either a nasopharyngeal swab or bronchoalveolar lavage, were retrospectively retrieved. Any patients with (i) a previous clinical history of lung disease; (ii) age < 18 years; or (iii) inadequate HRCT due to motion artifacts were excluded from the study [11,12,13,14]. Inclusion criteria were: (1) SARS-CoV-2 positive (on swab or bronchial wash); (2) age > 18 years and <80 years; (3) PaO_2_:FiO_2_ < 250 mmHg; (4) bilateral infiltrates at chest radiography; (5) CRP > 100 mg/L; and/or (6) diagnosis of acute respiratory distress syndrome (ARDS) according to the Berlin definition during hospitalization; and (7) at least one HRCT performed within 3 months after hospital discharge as an alternative to criteria (4) and (5) [11,12,13,14]. 

Upon admission to our hospital unit, all patients with severe COVID-19 received high-flow oxygen nasal cannula (HFNC) as initial safe standard oxygen treatment [15], passing then to either noninvasive or invasive mechanical ventilation (MV) if gas exchange worsened during HFNC. At the initial evaluation, all patients received noninvasive CPAP to assess the PF ratio, as suggested by the Berlin definition of ARDS [2]. The relevant demographic and clinical data were manually extrapolated from electronic medical records or charts and anonymously coded into a standardized data collection form. Two independent, experienced physicians assessed the data, and another researcher evaluated any differences in interpretation between the primary reviewers whenever necessary. HRCT was performed by a 256-row multidetector CT system (Brilliance iCT 256, Philips, Best, The Netherlands), and images were acquired during a single breath hold at full inspiration, with the patient in a supine position. The technical parameters were: rotation time, 270 ms; beam collimation, 128 × 2 × 0.625 mm; normalized pitch, 0.975; z-axis coverage, 160 mm; reconstruction interval, 0.3 mm; section reconstruction thickness, 1 mm; tube voltage, 120 kV; tube current (effective mA), 280–400 depending on patient size; and a 40 cm field of view. The CT images were analyzed at standard lung window settings (window level of −600 HU and window width of 1600 HU) and mediastinal window settings (window level 400 HU and window width 40 HU). 

The images were reviewed in consensus by two thoracic radiologists with 15 and 10 years of experience, respectively. Any discrepant interpretations were re-evaluated by a third specialist with 20 years’ experience. The following CT findings, as defined by the Fleischner Society glossary of terms, were evaluated [13]: ground glass opacities, air-space consolidations with or without air bronchogram, and the presence of signs of fibrosis (reticulations, traction bronchiectasis and bronchiolectasis, honeycombing, loss of volume, and/or architectural distortion). The loss of pulmonary volume was reported if fissures appeared to be retracted and abnormally misplaced [16]. Locations were described according to the pulmonary segmental subdivisions. If other relevant findings at HRCT were present, i.e., pleural effusion or pulmonary artery enlargement, they were also noted. The measurements of pulmonary artery diameter were obtained in the axial plane at the bifurcation of the pulmonary artery [17].

Spirometry was performed in the Pulmonology Units of the University Hospital of Trieste within one week before HRCT; the same machine was used for all patients. Global spirometry values were recorded. 

Continuous variables were summarized by mean and standard deviation or by median and range (minimum–maximum). The Shapiro–Wilk test was used to assess the distribution of numerical parameters, i.e. normality test. The data evidenced by the control CT scan were analyzed using absolute and percentage frequencies to identify radiographic abnormalities and determine their position in the lobes. The most common findings, i.e., ground glass, consolidations, reticulations, lobe volume reduction, bronchiectasis, bronchiolectasis, bands/opacity, and pleural effusion, were correlated with the results of the control spirometry. The Student’s t-test or the Mann–Whitney test assessed continuous variables, whilst categorical variables were compared by the Chi-square test of independence or by Fischer’s exact test, when appropriate, i.e., if more than 20% of the expected frequencies in the contingency table was <5 or at least one cell had an expected frequency of <1). The data analysis was made by the Software R (version 4.0.2, 2020), the statistical tests were two-tailed, and the level of statistical significance was set at a *p*-value of <0.05.

## 3. Results

Forty-four patients met the inclusion criteria and were included in the study. There were 32 males (72.7%) and 12 females (27.3%), and the average age was 64 (Standard Deviation [SD]: 12). A total of 22/44 patients were never smokers. All patients enrolled received medical therapy during hospitalization, i.e., 95.5% of them were put on prolonged low-dose methylprednisolone, in line with the protocol adopted in our Center for the treatment of ARDS [14]. Exposure to methylprednisolone complied with the following protocol: a loading dose of 80 mg intravenously (iv) at study entry (baseline), followed by an infusion of 80 mg/d in 240 mL of normal saline at 10 mL/h for at least 8 days, until achieving either a PaO_2_:FiO_2_ > 350 mmHg or a CRP < 20 mg/L; after which was performed oral administration at 16 mg or 20 mg iv twice daily until CRP reached <20% of the normal range or a PaO_2_:FiO_2_ > 400 (alternative SatHbO_2_ ≥ 95% on room air) [14].

A total of 43/44 patients required NIV (98.3%), and the clinical condition worsened in 12/44 (27.3%) who required IMV with pronation. The median time between the first positive RT-PCR and swab negativization was 23 days (range: 3–64), with a median hospitalization of 26.5 days (range: 14–194). Three patients died after hospital discharge (6.8%) due to ARDS-related complications. The clinical characteristics of the patients are reported in Table 1. 

### 3.1. HRCT Imaging Findings

The median time lapse between ARDS diagnosis and follow-up HRCT was 2.8 months, ranging from 1.9 to 3.7; the median time lapse between swab negativization and HRCT was 61.5 days (range: 17–168). Alterations in the pulmonary parenchyma were observed in 97.7% of patients. Linear bands were the most common finding (84%), followed by GGOs (75%), reticulations (34%), bronchiolectasis (32%), consolidations (30%), bronchiectasis (30%), and volume loss (25%) (Figure 1). The imaging data had a symmetric distribution, and the lower lobes were the most affected areas (Figure 2, Table 2).

A statistically significant increase in linear bands at the level of the right lung was observed in smokers compared to never-smokers (90.5% vs. 63.6%, respectively, *p*-value = 0.04) (Table 3). No other statistically significant values were observed (Table 3). A statistical significant presence of bronchiectasis was observed in patients who were invasively ventilated compared to those who were not (53.9% vs. 16.1%, respectively, *p*-value = 0.02) (Table 4). No other statistically significant values are observed (Table 4). A statistically significant increase in bronchiectasis and bronchiolectasis at the level of the right lung was observed in patients who were pronated compared to those who were not pronated (90.5% vs. 63.6%, respectively, *p*-value = 0.04) (Table 5). No other statistically significant values were observed (Table 5).

#### 3.1.1. Ground Glass Opacities

Ground glass opacities were observed in 33/44 patients (Figure 3). They were mainly in the right lung, particularly the posterior lower lobe, where all segments were similarly involved (40.9% at the apical segment, 56.8% in the posterior one). GGOs were present in the posterior segment of the upper lobe in 52.3% of cases, whilst the observation of GGOs in the two segments of the middle lobe and anterior and apical segments of the upper lobe was less common, i.e., 40.9%, 20.5%, 31.8%, and 29.5%, respectively. Similar findings were observed in the left lung, with a similar trend in all the lower lobe segments (from a minimum of 38.6% at the apical segment to 47.7% at the posterior one). The lingula was involved in about 40% of cases. The apicoposterior segment of the upper lobe was the most commonly involved segment (50%), whilst the anterior lobe was rarely involved (27.3%).

#### 3.1.2. Consolidations

Only 13/44 patients had parenchymal consolidations, 9 were bilateral and 4 were observed in the right lung (Figure 4). The most common sites were the posterior segments of the right upper lobe (18.2%), followed by the right lower lobe (15.9%). The middle lobe was rarely involved (2.3%). The parenchymal consolidation distribution was symmetrical in the left lung, with a prevalent involvement of the posterior segment (13.6%) of the lower lobe; lingula involvement was less common (6.8%). 

#### 3.1.3. Linear Bands

Linear bands were observed in 37/44 cases (84%). The inferior lobe was the most common site in the right lung, prevalently at the posterior segment (12 patients, 27.3%), followed the middle lobe (8 patients, 18.2%). Subpleural lines were more frequent in the posterior (18.2%) and lateral (15.9%) segments. Similar findings were observed in the left lung, with linear bands more commonly observed in the lower lobe, mostly at the posterior segments (11 patients, 25%) and at the inferior segment of the lingula (8 patients, 18.2%). Most of the subpleural lines were observed in the lateral segment of the inferior lobe (7 patients, 15.9%), followed by the anterior (6 patients, 13.6%) and posterior segments (6 patients, 13.6%) (Figure 5).

#### 3.1.4. Reticulations

A total of 15/44 patients (34%) had reticulations at imaging; 10 were bilateral, and 4 were observed in the right lung and 1 was observed in the left lobe. A total of 15/44 patients had reticulations in the right lung, most commonly at the upper and middle lobe, in a similar percentage of cases (18.2%). The lower lobe was also commonly involved, especially at the apical and posterior segments (22.7% each). A total of 11/44 (25%) patients had reticulations in the left lung, most commonly at the lingula (18.2%), followed by the posterior, lateral, and anterior segments of the lower lobe (15.9% each) (Figure 6).

#### 3.1.5. Bronchiectasis/Bronchiolectasis

A total of 27 patients had bronchiectasis (13/44) and bronchiolectasis (14/44). Bronchiectasis was observed in 8/13 patients (18.2%) in the posterior segment of the lower lobe of the right lung, closely followed by the middle lobe in 7/13 (16%) and the lateral segment of the inferior lobe in 6/13 (13.6%). Bronchiolectasis was more common in the posterior segment of the inferior lobe (9 patients, 20.5%). 

Bronchiectasis was observed in the left lung in 5/44 (11.4%) patients in the lateral and posterior segments of the lower lobe and the superior segment of the lingula. Like bronchiectasis, bronchiolectasis was more predominant in the posterior segment of the lower lobe of the left lung (7/14 patients, 16%). 

#### 3.1.6. Loss of Pulmonary Volume

Pulmonary volume loss was a less common finding, observed in 25% of cases. The right lung was affected in 10/44 patients (22.7%), most commonly involving the lower lobe (9/44), followed by the upper lobe (6/44). Loss of left pulmonary volume was observed in 11/44 patients (25%), with the lower lobe being the most common site (10/44 patients, 22.7%), like in the right lung (Figure 7).

#### 3.1.7. Other Findings

Other parenchymal features, including architectural distortion, subpleural bullae, cysts, nodules, and tree-in-bud opacities, were a rare finding. No honeycombing was observed in our study population, whilst pleural effusion was observed in 5/44 cases (11%). Moreover, no enlargement of the main pulmonary artery was identified, with a mean diameter of the main pulmonary artery (PA) of 27 mm (IQR 26–30) and a PA diameter-to-aorta ratio of 0.79 (IQR 0.72–0.91).

### 3.2. Correlation with Respiratory Function Tests

A total of 30/44 patients had a spirometry test during follow-up. The remaining 14 were either lost to follow-up or had difficulties in performing the examination that led to nondiagnostic results, which unfortunately may constitute a bias. However, spirometry data was available for 68.1% of them, and more than half had spirometry values within the normal limits; in 3/30 (10%) there was a mild obstructive alteration, and in 5/30 (16.7%) a mild restrictive pattern. A reduction of diffusing capacity for carbon monoxide (DLCO) was observed in 12/30 patients (40%). Presence of bronchiectasis and bronchiolectasis in both lungs was statistically significantly correlated with spirometry values (*p*-value = 0.009). Loss of pulmonary volume in the left lung also correlated with functional tests (*p*-value = 0.026), whilst this correlation is not present for the right lung (*p*-value = 0.12) (Table 6).

There was no correlation between the GGOs or consolidations and the spirometry values. However, it was noted that 4/6 patients (66.7%) with consolidations in the right lung had a reduced DLCO, whilst only 8/24 (33.3%) of patients with no consolidations had an altered DLCO. Similar findings were observed for reticulations, with reduced DLCO values in 4/6 (66.7%) patients with reticulations in the left lung, whilst only 2/6 (33.3%) of patients with no reticulations had an altered DLCO. 

The 5/6 patients (83.3%) who had loss of pulmonary volume and a DLCO reduction also had GGOs, bronchiectasis/bronchiolectasis, and linear bands at HRCT; only two of them had consolidations, and none had reticulations.

## 4. Discussion

This study details the short-term radiological findings at HRCT in patients affected by a severe SARS-CoV-2 infection with the development of ARDS during hospitalization. Literature reports that 70% of patients who recover from ARDS, whatever the underlying cause, have abnormal imaging findings at 6 months and in some cases may develop complications such as fibrosis and pulmonary hypertension [7]. Our study shows that also in COVID-19-related ARDS, 97.7% of patients had imaging documented sequelae at the follow-up CT scan. Noteworthy is the type of imaging findings observed in these patients after hospital discharge and the most common sites they occurred in. Indeed, the most common radiological findings were linear bands with variable paths, from indifferent to subpleural, ground-glass opacities, and, less commonly, also reticulations, observed in 84%, 75%, and 34% of patients, respectively. 

Literature has described the onset of fibrosis after ARDS, and it has mainly been related to the barotrauma induced by mechanical ventilation on the alveolar structures [18]. Moreover, some authors reported that previous coronavirus-related epidemic infections, such as severe acute respiratory syndrome (SARS) and the Middle East respiratory syndrome (MERS), are related to a higher incidence of fibrosis in 33% and 38% of patients at a three-month follow-up, respectively [19]. These preliminary findings suggest that attention should be paid to a possible postinfectious fibrotic evolution of COVID-19 and prompt further studies are needed to assess the longer-term outcomes in this selected population of patients with severe SARS-CoV-2 infection. 

The bilateral posterior lung segments were the most commonly involved areas in our series in both the upper and lower lobes. In addition to the most common patterns (GGOs and linear bands), bronchiectasis/bronchiolectasis and loss of volume were also present at this level. It is still not clear if the preferential involvement of posterior zones of the lung parenchyma is related to a direct virus-related parenchymal damage or to ventilation-related barotrauma. Indeed, it remains to be determined if the fibrotic outcome of SARS-CoV-2-related ARDS may have characteristics and presentations other than fibrosis secondary to ARDS due to other causes, where the related findings predominate in the anterior nondependent zones of the upper lobes [18]. 

Han et al. reported having observed that about one third of the patients treated with noninvasive ventilation developed fibrotic-like changes, in particular the elderly ones with a history of COVID-related ARDS [20]. However, these authors did not report on the location of the findings on the basis of an HRCT. Moreover, our series observed only mild fibrotic-like changes that included reticulations and a small percentage of bronchiectasis/bronchiolectasis and loss of volume.

Furthermore, Shan et al. reported in their study population that 65% of survivor patients 3 months after hospital discharge presented reticulations or traction bronchiectasis [21], while in our study population these pulmonary sequelae were reticulations in 34% and bronchiectasis in 30%, respectively. 

We can assume that the minor presence of these alterations in our population may be correlated with the use of glucocorticoids. In fact, other studies report benefits regarding symptoms and radiological picture in patients with post-COVID inflammatory lung diseases [22].

Literature reports that pleural effusion is an atypical finding in SARS-CoV2-2 infection [20,21,22,23,24,25,26,27,28,29,30]. Indeed, the presence of pleural effusion observed in 11% of our series could well be attributed to coexisting comorbidities rather than the infectious process itself, as it occurred in the elderly patients and those with underlying cardiovascular diseases. 

As to the secondary aim of this study (correlation between the HRCT findings and the respiratory functional tests at follow-up), we observed that the most common alteration at spirometry was a mild DLCO reduction (<80%, in 12/30 patients, 40%). This is in line with previously reported data during the SARS and MERS epidemics [24], and recent studies have reported a reduced DLCO in COVID-19 patients [25,26]. A study carried out in China in 2020 evaluated the presence of functional alterations in 55 patients who did not develop a severe pneumonia, 3 months after the resolution of the COVID-19 infection [27]. They also reported that a DLCO reduction (25.45% of patients) was the most frequent alteration. Although these authors reported a slightly lower percentage than ours, this discrepancy may be related to the different characteristics of our case series. That is, our study included critical patients with a more severe clinical onset where it is reasonable to expect a larger percentage of DLCO reduction.

The main limitations of this study are its retrospective nature and the small, although homogeneous, sample size, as well as the fact that not all patients were able to do the spirometry test. To the best of our knowledge, this is the first study to classify parenchymal abnormalities after SARS-CoV-2-related ARDS, according to segmental pulmonary anatomy. 

## 5. Conclusions

In conclusion, post-discharge HRCT of patients with a resolved severe SARS-CoV-2 infection complicated by ARDS demonstrated parenchymal alterations at short-term follow-up. The most common patterns (GGOs and linear bands), bronchiectasis/bronchiolectasis, and loss of volume were present in the bilateral posterior lung segments. Whilst pulmonary opacities varied in how quickly they gradually resolved, a long-term follow-up is a must to identify fibrotic-like changes to establish whether they are reversible, progressive, or permanent. To date, it cannot yet be determined whether these COVID-related fibrotic changes are reversible or not. Therefore, these patients should be given both a radiological and clinical follow-up to determine the most appropriate management and therapeutic strategies. Hopefully, further studies will provide the much needed data for a better understanding of the radiological and clinical evolution of SARS-CoV-2 severe pneumonia.

## Figures and Tables

**Figure 1 jcm-10-03985-f001:**
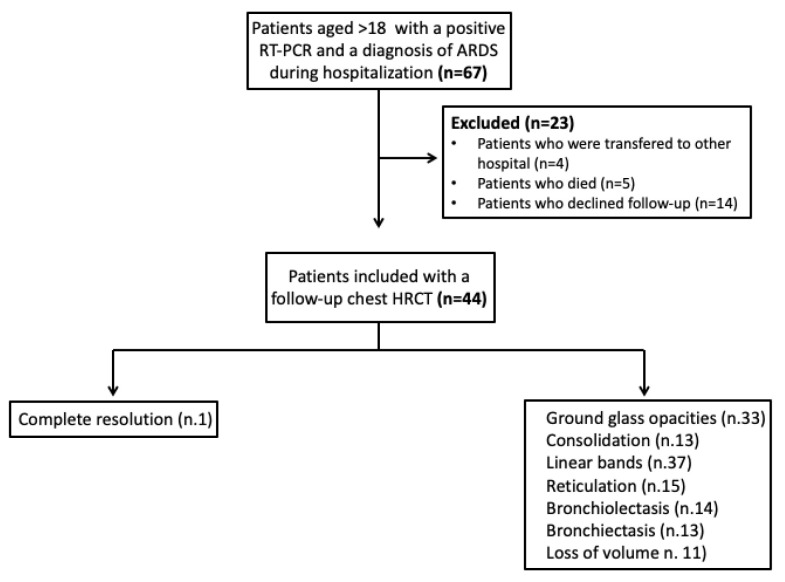
HRCT findings in the included population. Of 44 patients with COVID-related ARDS who underwent HRCT, only one had complete resolution, while the remaining patients had abnormal findings, most commonly linear bands, ground glass opacities, and reticulations.

**Figure 2 jcm-10-03985-f002:**
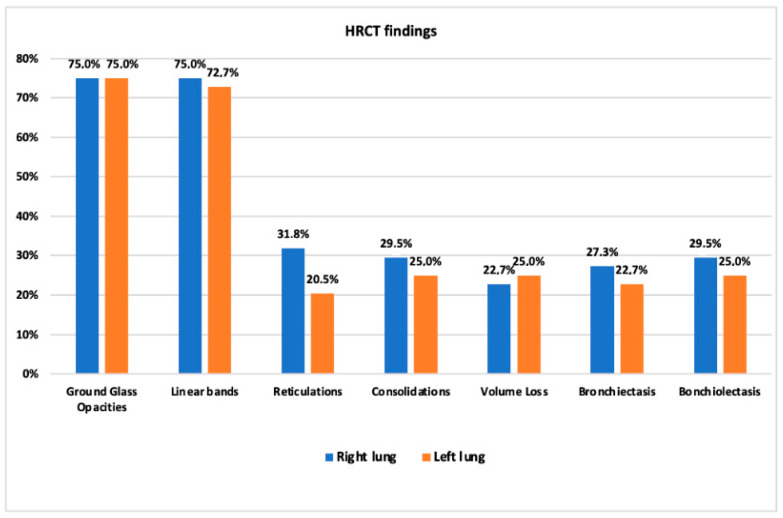
Incidence of HRCT findings divided by lung.

**Figure 3 jcm-10-03985-f003:**
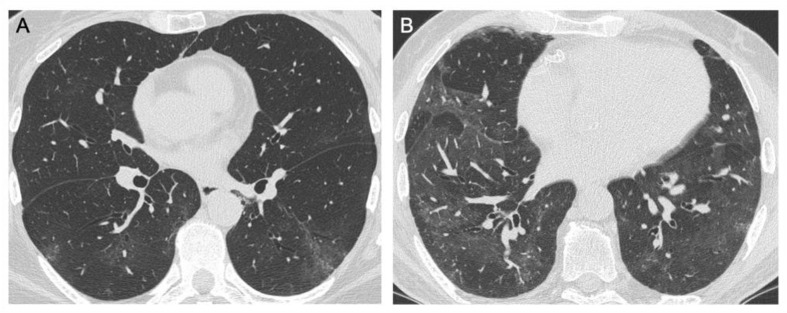
Axial HRCT images. (**A**) Ill-defined subpleural ground glass opacities in the lower lobes bilaterally; (**B**) Diffuse ground glass opacities.

**Figure 4 jcm-10-03985-f004:**
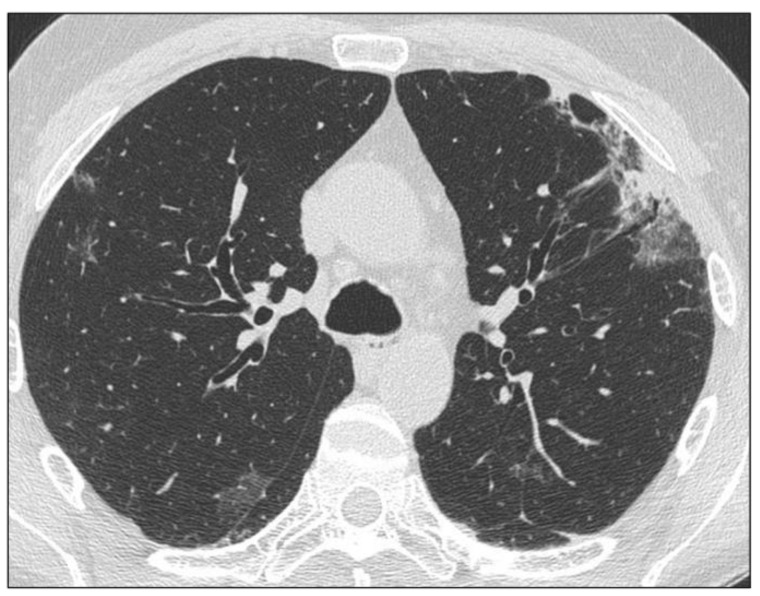
Axial HRCT image showing bilateral ground glass opacities associated to subpleural consolidations in the left lung.

**Figure 5 jcm-10-03985-f005:**
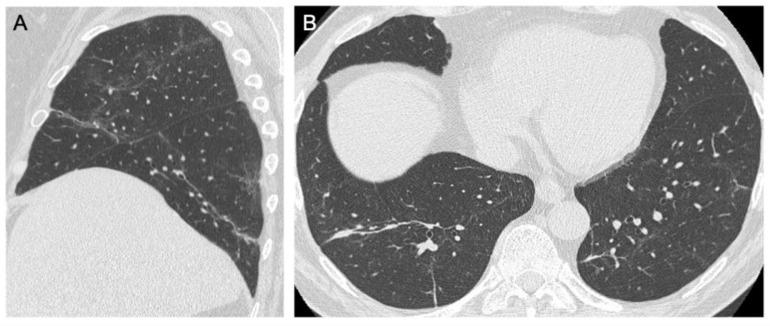
(**A**) HRCT multiplanar reconstruction on the sagittal plane of a 64-year-old woman in which ill-defined ground glass opacities and linear bands are observed. Spirometry values were normal; (**B**) Axial HRCT image of another patient showing only linear bands.

**Figure 6 jcm-10-03985-f006:**
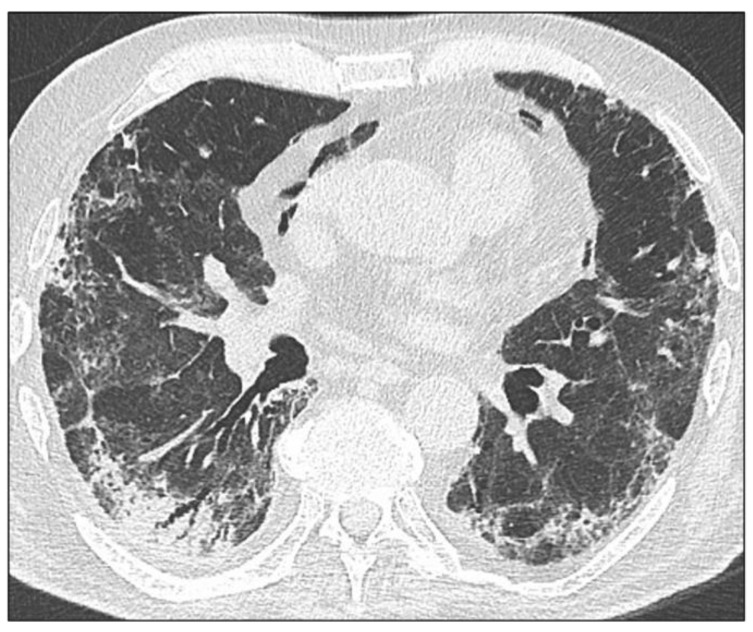
Axial HRCT image of a patient with extensive lung involvement, including diffuse reticulations, consolidations that predominate in the subpleural regions, and occurrence of pneumomediastinum.

**Figure 7 jcm-10-03985-f007:**
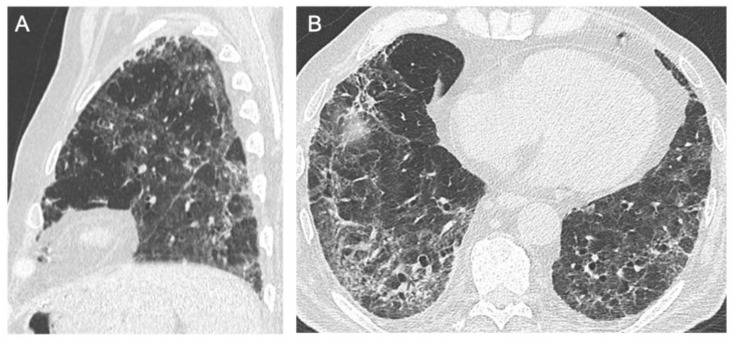
HRCT of a 72-year-old male who developed severe respiratory failure. The multiplanar reconstruction on the sagittal plane (**A**) shows loss of volume of the left lower lobe and diffuse ground glass opacities and consolidations; the axial plane (**B**) better demonstrates the presence of reticulations and bronchiectasis.

**Table 1 jcm-10-03985-t001:** Characteristics of the patients.

Characteristic	Patients (N = 44)
**Age (years)**	
Mean (SD)	64 (12)
**Gender (N,%)**	
Males	32 (72.7)
Females	12 (27.3)
**Smoking habit (N,%)**	
Never smoker	22 (51.2)
Yes, current smoker	11 (25.6)
Yes, former smoker	10 (23.3)
**Comorbidities (N,%)**	
None	11 (25.0)
One	14 (31.8)
≥2	19 (43.2)
**Type of Comorbidity (N,%)**	
Arterial hypertension	18 (40.9)
Diabetes	12 (27.3)
COPD	6 (13.6)
Cardiovascular diseases	9 (20.5)
Obesity	15 (34.1)
**PiO2:FiO2 (N,%)**	
<100	7 (16.3)
100–200	33 (76.7)
≥200	3 (7)
N/A	1
**Pharmacological treatments (N,%)**	
Corticosteroids	42 (95.5)
Antiviral agents	14 (31.8)
Hydroxychloroquine	20 (45.5)
Antibiotics	29 (65.9)
Remdesivir	0
Tocilizumab	4 (9.1)
**Intensive therapy (N,%)**	12 (27.3)
**NIV (N,%)**	43 (98.3)
**IMV (N,%)**	13 (29.5)
**Pronation (N,%)**	12 (27.3)
**Time between first positive swab and swab negativization (days)**	
Median (Min-Max)	23 (3–64)
**Hospitalization days**	
Median (Min-Max)	26.5 (14–194)
**Time between ARDS diagnosis and HRCT (months)**	
Median (Min-Max)	2.8 (1.9–3.7)
**Time between swab negativization and HRCT (days)**	
Median (Min-Max)	61.5 (17–168)
**Spirometry values (N/total number of patients, %)**	
Within normal limits	15/30 (50.0)
Mild obstructive deficit	3/30 (10.0)
Mild restrictive deficit	5/30 (16.7)
Reduction of DLCO	12/30 (40.0)
**Status of patients**	
Discharged	41 (93.2)
Deceased	3 (6.8)

NIV: noninvasive mechanical ventilation; IMV: invasive mechanical ventilation.

**Table 2 jcm-10-03985-t002:** Segmental distribution of the most common HRCT findings in each lung.

LEFT LUNG													
HRCT Finding	Upper Lobe				Lingula			Lower Lobe					
	Apicopost.		Ant.		Superior	Inferior		Superior		Anteromed.		Lat.	Post.
**Ground Glass Opacities (N,%)**	22 (50)		12 (27.3)		18 (40.9)	17 (38.6)		17 (38.6)		18 (40.9)		18 (40.9)	21 (47.7)
**Reticulations** **(N,%)**	5 (11.4)		6 (13.6)		8 (18.2)	8 (18.2)		6 (13.6)		7 (15.9)		7 (15.9)	7 (15.9)
**Linear bands,** **indifferent course (N,%)**	6 (13.6)		6 (13.6)		2 (4.5)	8 (18.2)		6 (13.6)		7 (15.9)		7 (15.9)	11 (25)
**Linear bands, subpleural (N,%)**	3 (6.8)		1 (2.3)		1 (2.3)	1 (2.3)		1 (2.3)		6 (13.6)		7 (15.9)	6 (13.6)
**Linear bands, perilobular (N,%)**	0		0		2 (4.5)	3 (6.8)		2 (4.5)		2 (4.5)		4 (9.1)	6 (13.6)
**Linear bands, subpleural and perilobular (N,%)**	0		0		0	0		1 (2.3)		2 (4.5)		3 (6.8)	3 (6.8)
**Bronchiectasis**	4 (9.1)		4 (9.1)		5 (11.4)	4 (9.1)		5 (11.4)		3 (6.8)		5 (11.4)	5 (11.4)
**Bronchiolectasis**	5 (11.4)		3 (6.8)		3 (6.8)	4 (9.1)		3 (6.8)		4 (9.1)		5 (11.4)	7 (15.9)
**RIGHT LUNG**													
**HRCT finding**	**Upper Lobe**				**Middle Lobe**			**Lower Lobe**					
	**Apical**	**Post.**		**Ant.**	**Lat.**		**Med.**	**Sup.**	**Ant.**		**Lat.**	**Post.**	**Med.**
**Ground Glass Opacities (N,%)**	13 (29.5)	23 (52.3)		14 (31.8)	18 (40.9)		9 (20.5)	18 (40.9)	19 (43.2)		23 (52.3)	25 (56.8)	18 (40.9)
**Reticulations (N,%)**	8 (18.2)	6 (13.6)		6 (13.6)	8 (18.2)		5 (11.4)	10 (22.7)	5 (11.4)		8 (18.2)	10 (22.7)	7 (15.9)
**Linear bands,** **Indifferent course (N,%)**	5 (11.4)	7 (15.9)		7 (15.9)	8 (18.2)		8 (18.2)	5 (11.4)	5 (11.4)		7 (15.9)	12 (27.3)	4 (9.1)
**Linear bands, subpleural (N,%)**	0	2 (4.5)		1 (2.3)	2 (4.5)		1 (2.3)	1 (2.3)	2 (4.5)		7 (15.9)	8 (18.2)	0
**Linear bands, perilobular (N,%)**	0	0		0	1 (2.3)		1 (2.3)	0	0		1 (2.3)	2 (4.5)	0
**Linear bands, subpleural and perilobular (N,%)**	0	1 (2.3)		0	3 (6.8)		1 (2.3)	2 (4.5)	3 (6.8)		5 (11.4)	5 (11.4)	0
**Bronchiectasis**	4 (9.1)	5 (11.4)		5 (11.4)	7 (15.9)		7 (15.9)	2 (4.5)	4 (9.1)		6 (13.6)	8 (18.2)	3 (6.8)
**Bronchiolectasis**	2 (4.5)	4 (9.1)		3 (6.8)	4 (9.1)		4 (9.1)	3 (6.8)	6 (13.6)		6 (13.6)	9 (20.5)	5 (11.4)

Apicopost.: apicoposterior; Ant.: anterior; Anteromed: anteromedial; Lat.: lateral; Post.: posterior; Med.: medial; Sup.: superior.

**Table 3 jcm-10-03985-t003:** Different HRTC patterns in smokers and never-smokers.

	Right Lung			Left Lung		
TC Patterns	Never Smokers (*n* = 22)	Smokers (*n* = 21)	*p*-Value	Never Smokers (*n* = 22)	Smokers (*n* = 21)	*p*-Value
**Ground Glass Opacities**	17 (77.3%)	16 (76.2%)	0.93	17 (77.3%)	16 (76.2%)	0.93
**Linear Bands**	**14 (63.6%)**	**19 (90.5%)**	**0.04**	15 (68.2%)	17 (86.0%)	0.34
**Reticulations**	8 (36.4%)	5 (23.8%)	0.37	5 (22.7%)	5 (23.8%)	0.93
**Consolidations**	6 (27.3%)	6 (28.6%)	0.92	4 (18.2%)	4 (19.1%)	0.94
**Loss of pulmunary Volume**	3 (13.6%)	6 (28.6%)	0.23	3 (13.6%)	7 (33.3%)	0.13
**Bronchiectasis**	5 (22.7%)	6 (28.6%)	0.66	3 (13.6%)	6 (28.6%)	0.23
**Bronchiolectasis**	4 (18.2%)	8 (38.1%)	0.15	4 (18.2%)	6 (28.6%)	0.42
**Total patterns** **Median (25p–75p)**	2 (1–4)	3 (2–5)	0.31	2 (1–3)	3 (1–5)	0.35
**Pulmonary artery diameter** **a.p. Median (25p–75p)**				27 (26–30)	27 (26–29)	0.79
**PA/AA ratio** **Median (25p–75p)**				0.77 (0.72–0.91)	0.79 (0.72–0.90)	0.86

a.p.: axial plane; PA: pulmonary artery; AA: ascending aorta.

**Table 4 jcm-10-03985-t004:** Different HRTC patterns in IMV and not-IMV.

	Right Lung				Left Lung	
TC Patterns	Not-IMV (*n* = 31)	IMV (*n* = 13)	*p*-Value	Not-IMV (*n* = 31)	IMV (*n* = 13)	*p*-Value
**Ground Glass Opacities**	24 (77.4%)	9 (69.2%)	0.57	24 (77.4%)	9 (69.2%)	0.57
**Linear Bands**	21 (67.7%)	12 (92.3%)	0.09	21 (67.7%)	11 (84.6%)	0.25
**Reticulations**	9 (29.0%)	5 (38.5%)	0.54	8 (5.8%)	3 (23.1%)	0.85
**Consolidations**	10 (32.3%)	3 (23.1%)	0.54	6 (19.4%)	3 (23.1%)	0.78
**Loss of pulmunary Volume**	5 (16.1%)	5 (38.5%)	0.11	6 (19.4%)	5 (38.5%)	0.18
**Bronchiectasis**	**5 (16.1%)**	**7 (53.9%)**	**0.02**	6 (19.4%)	4 (30.8%)	0.41
**Bronchiolectasis**	8 (25.8%)	5 (38.5%)	0.40	7 (22.6%)	4 (30.8%)	0.57
**Total patterns** **Median (25p–75p)**	2 (1–4)	4 (2–5)	0.13	2 (1–4)	2 (2–5)	0.39
**Pulmonary artery diameter** **a.p. Median (25p–75p)**				27 (26–29)	27 (25–31)	0.65
**PA/AA ratio** **Median (25p–75p)**				0.80 (0.73–0.91)	0.732 (0.63–0.84)	0.07

IMV: invasive mechanical ventilation; a.p.: axial plane; PA: pulmonary artery; AA: ascending aorta.

**Table 5 jcm-10-03985-t005:** Different HRTC patterns in pronated and not-pronated.

	Right Lung				Left Lung	
TC Patterns	No Pronated (*n* = 32)	Pronated (*n* = 12)	*p*-Value	No Pronated (*n* = 32)	Pronated (*n* = 12)	*p*-Value
**Ground Glass Opacities**	23 (71.9%)	10 (83.3%)	0.43	23 (71.9%)	10 (83.3%)	0.43
**Linear Bands**	23 (71.9%)	10 (83.3%)	0.43	23 (71.9%)	9 (75.0%)	0.84
**Reticulations**	8 (25.0%)	6 (50.0%)	0.11	6 (18.8%)	5 (41.7%)	0.12
**Consolidations**	8 (25.0%)	5 (41.7%)	0.28	6 (18.8%)	3 (25.0%)	0.65
**Loss of pulmunary Volume**	7 (21.9%)	3 (25.0%)	0.83	8 (25.0%)	3 (25.0%)	0.99
**Bronchiectasis**	**6 (18.8%)**	**6 (50.0%)**	**0.04**	6 (18.8%)	4 (33.3%)	0.30
**Bronchiolectasis**	**6 (18.8%)**	**7 (58.3%)**	**0.02**	6 (18.8%)	5 (42.7%)	0.12
**Total patterns** **Median (25p–75p)**	**2 (1–3.5)**	**4.5 (2.5–5)**	**0.03**	2 (1–3)	3 (2–5)	0.18
**Pulmonary artery diameter** **a.p. Median (25p–75p)**				**27 (25–29)**	**30 (27–31.5)**	**0.03**
**PA/AA ratio** **Median (25p–75p)**				0.76 (0.71–0.88)	0.88 (0.79–0.94)	0.06

a.p.: axial plane; PA: pulmonary artery; AA: ascending aorta.

**Table 6 jcm-10-03985-t006:** Correlation between radiological findings and spirometry values.

RIGHT LUNG	
Main HRCT findings	Spirometry values
Ground Glass Opacities	*p* = 0.36
Consolidations	*p* = 0.18
Reticulations	*p* = 0.66
Lobar volume reduction	*p* = 0.12
**Bronchiectasis**	***p*** **= 0.009**
**Bronchiolectasis**	***p*** **= 0.0123**
Linear Bands	*p* = 0.09
**LEFT LUNG**	
**Main HRCT findings**	**Spirometry values**
Ground Glass Opacities	*p* = 0.36
Consolidations	*p* = 0.27
Reticulations	*p* = 0.18
**Lobar volume reduction**	***p* = 0.026**
**Bronchiectasis**	***p* = 0.002**
**Bronchiolectasis**	***p* = 0.03**
Linear Bands	*p* = 0.67

## Data Availability

All the data are available upon reasonable request to the corresponding author.

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
