# Peer review of "Interstitial Lung Disease at High Resolution CT after SARS-CoV-2-Related Acute Respiratory Distress Syndrome According to Pulmonary Segmental Anatomy"

_jcm, 2021, doi:10.3390/jcm10173985_

Round 1
Reviewer 1 Report
The authors described in a detailed way CT findings in a group of patients who survived SARS-CoV-2- related ARDS. The paper is well written and easy to understand.
Despite this, some aspects need to be pointed out in a clearer way to make the paper more interesting and original:
Minor comments:
- All the tables should be adjusted in a clearer way from a graphical point of view
- There are some minor mistakes to check in the text (e.g. "lobe" instead of "lung" page 9 line 200 and line 201)
Major comments:
- Half of the patients were smokers (current or former). Have you investigated if this habit was correlated with any particular radiological pattern? There were some peculiar differences between smoker and never smoker patients after COVID-19 related ARDS?
- At the same time, it would be interesting to know if a particular radiological pattern was found in patients who were invasively ventilated compared to those who were not.
- The description of the different radiological patterns is reported in a detailed way according to the lung segmental distribution but an interesting point, in my opinion, should be the comparison with the CTs performed during the acute phase of ARDS to evaluate how the CT findings have evolved over the time. Do the different CT findings reported in the lung segments reflect the presence of more compromised area during the acute phase?
- Concerning the correlation between the HRCT findings and the respiratory functional tests at follow up it is an interesting point but to understand the real decrease in functional respiratory tests and its correlation with CT, it should be compared with the baseline values for each patient (before COVID infection). Did you perform this analysis?
- To conclude, it is not so clear to me what is the central message of the discussion. What is the authors' point of view/hypotesis regarding the different localization of the different CT findings in these patients? This should be discussed.
Author Response
A) Minor comments:
- All the tables should be adjusted in a clearer way from a graphical point of view
R: We would like to thank the reviewer for these comments that have enabled us to enhance our manuscript. As requested, the tables have been adjusted.
- There are some minor mistakes to check in the text (e.g. "lobe" instead of "lung" page 9 line 200 and line 201)
R: We would like to thank the reviewer for this comment. As requested, the minor mistakes have been corrected.
B) Major comments:
1.Half of the patients were smokers (current or former). Have you investigated if this habit was correlated with any particular radiological pattern? There were some peculiar differences between smoker and never smoker patients after COVID-19 related ARDS?
R: We thank the reviewer for this comment. As requested, we investigated the correlation between smoker habit with any particular radiological pattern in our patients and we observed significant pattern differences i.e. “linear bands”. We added this sentence and table 4 to the Manuscript: (Results. 3.1. HRCT imaging findings) ”A statistically significant increase in linear bands at the level of right lung was observed in smokers compared to never-smokers (90.5% vs 63.6% respectively, p-value=0.04) (Table 3). No other statistically significant values were observed (Table 3)”.
2.At the same time, it would be interesting to know if a particular radiological pattern was found in patients who were invasively ventilated compared to those who were not.
R: We thank the reviewer for this comment. As requested, we investigated if a particular radiological pattern was present in patients who were invasively ventilated compared to those who were not. A statistically significantly higher presence of bronchiectasis was observed in patients who were invasively ventilated. We added this sentence (Results. 3.1. HRCT imaging findings) and table 5 to the Manuscript: “A statistical significant presence of bronchiectasis was observed in patients who were invasively ventilated compared to those who were not (53.9% vs 16.1% respectively, p-value =0.02) (Table 4). No other statistically significant values are observed (Table 4)”.
3.The description of the different radiological patterns is reported in a detailed way according to the lung segmental distribution but an interesting point, in my opinion, should be the comparison with the CTs performed during the acute phase of ARDS to evaluate how the CT findings have evolved over the time. Do the different CT findings reported in the lung segments reflect the presence of more compromised area during the acute phase?
R: We thank the reviewer for this comment. During the first wave, given the large number of patients with severe pneumonia due to COVID-19, only chest X-rays were routinely performed in our hospital. Therefore, unfortunately, we cannot make this comparison.
4.Concerning the correlation between the HRCT findings and the respiratory functional tests at follow up it is an interesting point but to understand the real decrease in functional respiratory tests and its correlation with CT, it should be compared with the baseline values for each patient (before COVID infection). Did you perform this analysis?
R: We thank the reviewer for this comment. As requested, we clarified this in the Manuscript (2. Materials and Methods): “Any patients with (i) a previous clinical history of lung disease were excluded from the study”.
- To conclude, it is not so clear to me what is the central message of the discussion. What is the authors' point of view/hypotesis regarding the different localization of the different CT findings in these patients? This should be discussed.
R: We thank the reviewer for this comment. We hypothesize that our treatment protocol, using low-prolonged dose of methylprednisolone during the acute phase of ARDS, (14. Salton F, Confalonieri P, Meduri GU, et al 2020), prolonged Low-Dose Methylprednisolone in Patients With Severe COVID-19 Pneumonia. Open Forum Infect Dis 7:ofaa421. https://doi.org/10.1093/ofid/ofaa421, may prevent the development of pulmonary fibrosis in most patients, particularly peripheral and posterior lung fibrosis. The glucocorticoids might reduce the inflammatory burden and facilitate lung regeneration and healing (5. Meduri GU, Annane D, Confalonieri M, et al (2020) Pharmacological principles guiding prolonged glucocorticoid treatment in ARDS. Intensive Care Med 46:2284–2296. https://doi.org/10.1007/s00134-020-06289-8). Further studies are ongoing to support these preliminary data. We have improved the conclusion: “In conclusion, post-discharge HRCT of patients with a resolved severe SARS-CoV-2 infection, complicated by ARDS demonstrated parenchymal alterations at short-term follow-up. The most common patterns (GGOs and linear bands), bronchiectasis/bronchiolectasis and loss of volume were present in the bilateral posterior lung segments. Whilst pulmonary opacities varied in how quickly they gradually resolved, a long-term follow-up is a must to identify fibrotic-like changes to establish whether they are reversible, progressive or permanent. As to date it cannot yet be determined whether these Covid-related fibrotic changes are reversible or not. Therefore, these patients should be given both a radiological and clinical follow-up to determine the most appropriate management and therapeutic strategies. Hopefully, further studies will provide the much needed data for a better understanding of the radiological and clinical evolution of SARS-CoV-2 severe pneumonia”.
Reviewer 2 Report
The authors describe their single center experience with follow up imaging obtained in COVID-19 survivors who had been hospitalized for acute hypoxic respiratory syndrome and had received treatment with corticosteroids. Given the significantly higher use of corticosteroids in COVID-19 patients with respiratory failure compared with ARDS at large, this is useful data to report. I have several comments/considerations to suggest:
Major:
- A comparison with age-matched COVID patients who did not receive steroids would be helpful, particularly to support the assertion that the lower incidence of persistent abnormalities is due to steroid use.
- ARDS is generally defined in patients receiving mechanical ventilation. A significant proportion of patients did not undergo mechanical ventilation. Furthermore, validation that patients had ARDS (by description of initial imaging) is not reported.
- Correlation between initial imaging findings from hospitalization and follow up imaging would be helpful.
- Correlation between specific PFT abnormalities and imaging abnormalities would be helpful.
- How was volume loss defined? this is not described in the methods. More quantitative analysis of imaging could be of interest, but may not fit within the scope of the manuscript (with the possible exception of lung volumes and pulmonary artery sizes).
- What was the usual treatment course of corticosteroids?
- I am not sure what the relevance is of the specific segmental anatomic distribution described. If there is a hypothesis meriting its inclusion, please state so. Otherwise, would consolidate, eliminate, or place this section in supplement.
Minor:
- What does BPCO in the Table 1 stand for? This table would benefit from more compact formatting as well.
- Did some patients who underwent NIV get high flow nasal cannula or did all receive non-invasive positive pressure ventilation?
Author Response
Major:
1.A comparison with age-matched COVID patients who did not receive steroids would be helpful, particularly to support the assertion that the lower incidence of persistent abnormalities is due to steroid use.
R: We would like to thank the reviewer for these comments that have enabled us to enhance our manuscript.
This is an excellent observation for further articles because in this study almost all of our patients received corticosteroids according to the published protocol (Salton F, Confalonieri P, Meduri GU, et al (2020) Prolonged Low-Dose Methylprednisolone in Patients With Severe COVID-19 Pneumonia. Open Forum Infect Dis 7:ofaa421). We emphasised this information in the manuscript: “All patients enrolled received medical therapy during hospitalization, i.e., 95.5% of them were put on prolonged low-dose methylprednisolone, in line with the protocol adopted in our Center for the treatment of ARDS [14]. Exposure to methylprednisolone complied with the following protocol: a loading dose of 80 mg intravenously (iv) at study entry (baseline), followed by an infusion of 80 mg/d in 240 mL of normal saline at 10 mL/h for at least 8 days, until achieving either a PaO2:FiO2 >350 mmHg or a CRP <20 mg/L; after which, oral administration at 16 mg or 20 mg iv twice daily until CRP reached <20% of the normal range or a PaO2:FiO2 >400 (alternative SatHbO2 ≥95% on room air) [14]”.
2.ARDS is generally defined in patients receiving mechanical ventilation. A significant proportion of patients did not undergo mechanical ventilation. Furthermore, validation that patients had ARDS (by description of initial imaging) is not reported.
R: We thank the reviewer for this comment. As requested, more details have been included as follows (2. Materials and Methods): “Any patients with (i) a previous clinical history of lung disease; (ii) age < 18 years; (iii) inadequate HRCT due to motion artifacts, were excluded from the study [11-14]. Inclusion criteria were: (1) SARS-CoV-2 positive (on swab or bronchial wash); (2) age >18 and <80 years; (3) PaO2:FiO2 <250 mmHg; (4) bilateral infiltrates at chest radiography; (5) CRP >100 mg/L; and/or 6) diagnosis of acute respiratory distress syndrome (ARDS) according to the Berlin definition during hospitalization, 7) at least one HRCT performed within 3 months after hospital discharge as an alternative to criteria (4) and (5) [11-14]. Upon entering the hospital our hospital unit, all patients received high-flow oxygen nasal cannula (HFNC) as initial safe standard oxygen treatment in our patients with severe COVID-19 [15] passing then to mechanical ventilation (MV), either noninvasive or invasive MV, if gas exchange worsened during HFNC. At the initial evaluation all the patients received noninvasively CPAP to assess the ratio P:F as requested by the Berlin definition of ARDS [2]”.
3.Correlation between initial imaging findings from hospitalization and follow up imaging would be helpful.
R: We thank the reviewer for this comment. During the first wave, given the large number of patients with severe pneumonia due to COVID-19, only chest x-rays were routinely performed in our hospital. Therefore, unfortunately, we cannot make this comparison.
4.Correlation between specific PFT abnormalities and imaging abnormalities would be helpful.
R: We thank the reviewer for this comment. We emphatised this correlation in Table 6 and in the paragraph (3.2. Correlation with respiratory function tests): ” A total of 30/44 patients had a spirometry test during follow-up. The remaining 14 were either lost to follow-up, or had difficulties in performing the examination, that led to non-diagnostic results, which, unfortunately may constitute a bias. However, spirometry data was available for 68.1% of them and more than half had spirometry values within the normal limits; in 3/30 (10%) there was a mild obstructive alteration and in 5/30 (16.7%) a mild restrictive pattern. A reduction of Diffusing Capacity for Carbon Monoxide (DLCO) was observed in 12/30 patients (40%). Presence of bronchiectasis and bronchiolectasis in both lungs was statistically significantly correlated with spirometry values (p-value=0.009). Loss of pulmonary volume in the left lung also correlated with functional tests (p-value=0.026), whilst this correlation is not present for the right lung (p-value=0.12) (Table 6).
There was no correlation between the GGOs or consolidations and the spirometry values. However, it was noted that 4/6 patients (66.7%) with consolidations in the right lung had a reduced DLCO, whilst only 8/24 (33.3%) of patients with no consolidations had an altered DLCO. Similar findings were observed for reticulations, with reduced DLCO values in 4/6 (66.7%) patients with reticulations in the left lung, whilst only 2/6 (33.3%) of patients with no reticulations had an altered DLCO. The 5/6 patients (83.3%), who had loss of pulmonary volume and a DLCO reduction, also had GGOs, bronchiectasis/bronchiolectasis and linear bands at HRCT; only two of them had consolidations and none had reticulations”.
5.How was volume loss defined? this is not described in the methods. More quantitative analysis of imaging could be of interest, but may not fit within the scope of the manuscript (with the possible exception of lung volumes and pulmonary artery sizes).
R: We thank the reviewer for this comment. We noted this correlation in these two sentences:
“The loss of pulmonary volume was reported if fissures appeared to be retracted and abnormally misplaced [16]”
“The measurements of pulmonary artery diameter were obtained in the axial plane at the bifurcation of the pulmonary artery [17]”; “Moreover no enlargement of the main pulmonary artery was identified, with a mean diameter of main pulmonary artery of 27 mm (IQR 26-30) and a PA diameter-to-aorta ratio of 0.79 (IQR 0.72-0.91)”.
6.What was the usual treatment course of corticosteroids?
R: We would like to thank the reviewer for this comment that has enabled us to enhance our manuscript. We noted this information in the manuscript (3. Results): “All patients enrolled received medical therapy during hospitalization, i.e., 95.5% of them were put on prolonged low-dose methylprednisolone, in line with the protocol adopted in our Center for the treatment of ARDS [14]. Exposure to methylprednisolone complied with the following protocol: a loading dose of 80 mg intravenously (iv) at study entry (baseline), followed by an infusion of 80 mg/d in 240 mL of normal saline at 10 mL/h for at least 8 days, until achieving either a PaO2:FiO2 >350 mmHg or a CRP <20 mg/L; after which, oral administration at 16 mg or 20 mg iv twice daily until CRP reached <20% of the normal range or a PaO2:FiO2 >400 (alternative SatHbO2 ≥95% on room air) [14]”.
7.I am not sure what the relevance is of the specific segmental anatomic distribution described. If there is a hypothesis meriting its inclusion, please state so. Otherwise, would consolidate, eliminate, or place this section in supplement.
R: We would like to thank the reviewer for this comment. The objective of our study was to evaluate the distribution of parenchymal changes do to post-COVID-19 ARDS and to describe the specific segmental anatomic distribution to focus the reader's attention in these areas that literature has not reported as being involved in ARDS do to other causes. We improved the conclusion: “In conclusion, post-discharge HRCT of patients with a resolved severe SARS-CoV-2 infection, complicated by ARDS demonstrated parenchymal alterations at short-term follow-up. The most common patterns (GGOs and linear bands), bronchiectasis/bronchiolectasis and loss of volume were present in the bilateral posterior lung segments. Whilst pulmonary opacities varied in how quickly they gradually resolved, a long-term follow-up is a must to identify fibrotic-like changes to establish whether they are reversible, progressive or permanent. As to date it cannot yet be determined whether these Covid-related fibrotic changes are reversible or not. Therefore, these patients should be given both a radiological and clinical follow-up to determine the most appropriate management and therapeutic strategies. Hopefully, further studies will provide the much needed data for a better understanding of the radiological and clinical evolution of SARS-CoV-2 severe pneumonia”.
Minor:
1.What does BPCO in the Table 1 stand for? This table would benefit from more compact formatting as well.
R: We would like to thank the reviewer for this comment. We have done as requested.
2.Did some patients who underwent NIV get high flow nasal cannula or did all receive non-invasive positive pressure ventilation?
R: We would like to thank the reviewer for this comment. As requested, we added this paragraph (2. Materials and Methods) to the manuscript: “Upon admission to our hospital unit, all patients with severe COVID-19 received high-flow oxygen nasal cannula (HFNC) as initial safe standard oxygen treatment (Demoule A, Vieillarde Baron A, Darmon M, et al. High-Flow Nasal Cannula in Critically III Patients with Severe COVID-19. Am J Respir Crit Care Med 2020; 202: 1039-42) passing then to mechanical ventilation (MV), either noninvasive or invasive MV, if gas exchange worsened during HFNC”.

Reviewer 3 Report
Dear Editor and Authors,
I would like to thank you for requesting that I review this work titled “Interstitial Lung Disease at High Resolution CT after SARS-CoV-2-Related Acute Respiratory Distress Syndrome According to Pulmonary Segmental Anatomy” by Dr. Baratella and colleagues from Cattinara Hospital and the University of Trieste in Trieste, Italy.
As a thoracic surgeon who is now seconded in the Covid Ward and ICU the subject was quite of interest to me. The question of post Covid disease sequelae and the retaining lung damage in recuperated patients is still unanswered. As the authors also report, most patients post Covid infection return to normal lung function and imaging, however a significant percentage have persistent interstitial infiltrates, pulmonary fibrosis and a restrictive respiratory pattern. The truth is we do not know yet which patients will do so and which factors are associated with persistent lung damage. The authors in this retrospective case control analysis attempt to describe the high-resolution CT (HRCT) findings in patients post severe COVID-19 pneumonia complicated by ARDS and treated with systemic corticosteroid.
This is one of a number of similar studies based on the so far available follow up of these patients. It is a retrospective analysis, presumably of medical charts which by itself I feel is problematic in regards to completeness of data and information. Unfortunately during the pandemic one of the first things to “take second place” is accurate and constant documentation! Did the authors have adequate information for all the patients?
The language and structure of the manuscript is understandable and acceptable so I do not have anything to comment about it.
I do have some additional suggestions and comments for improvement.
Comments:
- The definition of ARDS based on reference 2 utilized the concept of PEEP which can easily misunderstood to pertain only to intubated patients. However, as reference 2’s consensus statement states “a minimum level of PEEP (5 cm H2O), which can be delivered non-invasively in mild ARDS, was included in the draft definition of ARDS”. The authors should clarify their statement.
- Line 72: the word recommendations is misspelled!
- Of the patients hospitalized in the Respiratory High Dependency Unit most required high-flow oxygen therapy and some where intubated and even pronated! Was there a difference in HRCT outcome between these two, presumably disparate in intensity of disease, groups?
- Its sample size is relatively small at 44 patients and in addition there seems to be a significant unrecognized bias as one of the inclusion criteria is the performance of a HRCT within 3 months post discharge. Also where HRCT performed in a single institution and machine? It is common, especially during the pandemic, if patients have private insurance to visit non-governmental facilities (where the perceived risk of Covid-19 infection is high) to undergo examinations!
- The three patients that died post discharge of ARDS related complications should they not be removed from the analysis sample as they clearly represent outliers?
- There is a wide disparity between the periods of HRCT performance from 2 months to double that at 4 months. This is a significant time period. Did the authors see better outcomes in patients who had their scan later on?
- There is no attempt of quantification of the extent/percentage of residual disease as compared to the original presentation and this is a significant limitation of the study. A patient who had significant opacities and ground glass to 40-50% of their lung and on follow up HCRT had a residual 10% is categorized as opacities even though there is clearly an improvement!
- Table 2’s designation needs to actually be changed to Figure 2 and needs to be corrected. Figures 2, 3, 4, 5 and 6 (all the Chest CT images) are superfluous. We all are aware how ground glass, opacification and bands appear on a scan!
- There were only 30 patients who were able to undergo spirometry evaluation. A significant portion where lost to follow up, died or where unable to perform the examination! Therefore, this further weakens the sample size mentioned earlier and the results of the analysis!
- There is no term “approached statistical significance” in line 245. There is only “there was or there wasn’t” statistical significance and definitively with a p-value of 0.12 it certainly was not “approaching significance”!
- In the discussion, which by the way is a bit longwinded and could benefit from a small reduction in size, there is no evidence presented by the authors that “these alterations in our population may 296 be correlated with the use of glucocorticoids”. Therefore this whole premise and statement need to be revised - removed.
In conclusion, this is a relatively small, retrospective study which in truth it is not offering much to elucidate the effect and sequelae of Covid pneumonia in recovering patients, particularly since there is no quantification of original to post disease findings! I wish all good health and good luck to the authors.
Author Response
I do have some additional suggestions and comments for improvement.
R: We would like to thank the reviewer for these comments that have enabled us to enhance our manuscript.
Comments:
1.The definition of ARDS based on reference 2 utilized the concept of PEEP which can easily misunderstood to pertain only to intubated patients. However, as reference 2’s consensus statement states “a minimum level of PEEP (5 cm H2O), which can be delivered non-invasively in mild ARDS, was included in the draft definition of ARDS”. The authors should clarify their statement.
R: We thank the reviewer for this comment. As requested, we substituted the term PEEP with CPAP in the manuscript . This sentence was added (2. Materials and Methods) :” At the initial evaluation, all patients received noninvasive CPAP to assess the PF ratio, as suggested by the Berlin definition of ARDS [2]”.
2.Line 72: the word recommendations is misspelled!
R: We thank the reviewer for this comment. As requested, the typo has been corrected.
3.Of the patients hospitalized in the Respiratory High Dependency Unit most required high-flow oxygen therapy and some where intubated and even pronated! Was there a difference in HRCT outcome between these two, presumably disparate in intensity of disease, groups?
R: We thank the reviewer for this comment. As requested, we added information as to this to the manuscript (3.1. HRCT imaging findings) and in table 5:”A statistically significant increase in bronchiectasis and bronchiolectasis at the level of right lung was observed in pronated compared to not-pronated (90.5% vs 63.6% respectively, p-value=0.04) (Table 5). No other statistically significant values were observed (Table 5)”.
4.Its sample size is relatively small at 44 patients and in addition there seems to be a significant unrecognized bias as one of the inclusion criteria is the performance of a HRCT within 3 months post discharge. Also where HRCT performed in a single institution and machine? It is common, especially during the pandemic, if patients have private insurance to visit non-governmental facilities (where the perceived risk of Covid-19 infection is high) to undergo examinations!
R: We thank the reviewer for this comment. The relatively small number of patients was also due to the fact that all HRCT were performed in a single University Center and all patients were evaluated by the same HRTC machine
5.The three patients that died post discharge of ARDS related complications should they not be removed from the analysis sample as they clearly represent outliers?
R: We thank the reviewer for this comment. The three patients are part of the sample because they met the inclusion criteria. Moreover, the 3 patients died after the end of the study observation period.
6.There is a wide disparity between the periods of HRCT performance from 2 months to double that at 4 months. This is a significant time period. Did the authors see better outcomes in patients who had their scan later on?
R: We thank the reviewer for this comment. To date we have not observed any significant differences but further studies are on going to support these results.
7.There is no attempt of quantification of the extent/percentage of residual disease as compared to the original presentation and this is a significant limitation of the study. A patient who had significant opacities and ground glass to 40-50% of their lung and on follow up HCRT had a residual 10% is categorized as opacities even though there is clearly an improvement!
R: We thank the reviewer for this comment. During the first wave, given the large number of patients with severe pneumonia due to COVID-19, only chest X-rays were routinely performed in our hospital. Therefore, unfortunately, we cannot make this comparison.
8.Table 2’s designation needs to actually be changed to Figure 2 and needs to be corrected. Figures 2, 3, 4, 5 and 6 (all the Chest CT images) are superfluous. We all are aware how ground glass, opacification and bands appear on a scan!
R: We thank the reviewer for this comment. We have done as requested.
We have included several images because we think they may be useful for the less experienced clinicians for use in patient follow-up to identify different HRTC patterns.
9.There were only 30 patients who were able to undergo spirometry evaluation. A significant portion where lost to follow up, died or where unable to perform the examination! Therefore, this further weakens the sample size mentioned earlier and the results of the analysis!
R: We thank the reviewer for this comment. As reported by thoracic surgeons in this war against SARS COV-2 has been difficult to hospitalize, treat and follow all patients. Furthermore, the evaluation of the correlation between the HRTC pattern and the spirometry evaluation was our second objective. We emphasised the the correlation between HRTC and spirometry evaluation in Table 6 and in the paragraph (3.2. Correlation with respiratory function tests): ” A total of 30/44 patients had a spirometry test during follow-up. The remaining 14 were either lost to follow-up, or had difficulties in performing the examination, that led to non-diagnostic results, which, unfortunately may constitute a bias. However, spirometry data was available for 68.1% of them and more than half had spirometry values within the normal limits; in 3/30 (10%) there was a mild obstructive alteration and in 5/30 (16.7%) a mild restrictive pattern. A reduction of Diffusing Capacity for Carbon Monoxide (DLCO) was observed in 12/30 patients (40%). Presence of bronchiectasis and bronchiolectasis in both lungs was statistically significantly correlated with spirometry values (p-value=0.009). Loss of pulmonary volume in the left lung also correlated with functional tests (p-value=0.026), whilst this correlation is not present for the right lung (p-value=0.12) (Table 6).
There was no correlation between the GGOs or consolidations and the spirometry values. However, it was noted that 4/6 patients (66.7%) with consolidations in the right lung had a reduced DLCO, whilst only 8/24 (33.3%) of patients with no consolidations had an altered DLCO. Similar findings were observed for reticulations, with reduced DLCO values in 4/6 (66.7%) patients with reticulations in the left lung, whilst only 2/6 (33.3%) of patients with no reticulations had an altered DLCO. The 5/6 patients (83.3%), who had loss of pulmonary volume and a DLCO reduction, also had GGOs, bronchiectasis/bronchiolectasis and linear bands at HRCT; only two of them had consolidations and none had reticulations”.
10.There is no term “approached statistical significance” in line 245. There is only “there was or there wasn’t” statistical significance and definitively with a p-value of 0.12 it certainly was not “approaching significance”!
R: We thank the reviewer for this comment. We have corrected this.
11.In the discussion, which by the way is a bit long winded and could benefit from a small reduction in size, there is no evidence presented by the authors that “these alterations in our population may 296 be correlated with the use of glucocorticoids”. Therefore this whole premise and statement need to be revised - removed.
R: We thank the reviewer for this comment. We hypothesize that our treatmentprotocol using low-prolonged dose of methylprednisoloneduring the acute phase of ARDS (Salton F, Confalonieri P, Meduri GU, et al (2020) Prolonged Low-Dose Methylprednisolone in Patients With Severe COVID-19 Pneumonia. Open Forum Infect Dis 7:ofaa421. https://doi.org/10.1093/ofid/ofaa421) may prevent the development of pulmonary fibrosis in most patients, particularly peripheral and posterior lung fibrosis. The glucocorticoids might reduce the inflammatory burden and facilitate lung regeneration and healing (Meduri GU, Annane D, Confalonieri M, et al (2020) Pharmacological principles guiding prolonged glucocorticoid treatment in ARDS. Intensive Care Med 46:2284–2296. https://doi.org/10.1007/s00134-020-06289-8). Further studies are ongoing to support these preliminary data.
Furthermore, our observational study, on a selected population, emphasize which lung areas the parenchymal changes were most frequently localized in at follow-up.

Round 2
Reviewer 1 Report
Thank you for your clarification. I would recommend pubblicato.
Reviewer 3 Report
Dear Editor and Authors,
I had the oportunity to re-read and re-evaluate the edited version of this manuscript. I also reviewed the submitted letter response of the authors to my comments. Although, I am not satisfied 100% by their responses, I do concede that they have made a conscious effort to modify their work, providing additional evidence/data, responding to the queries and making the majority of the changes/edits suggested. As such I am content to now recommend the publication of this work. I don’t feel it is “ground breaking” research but it may serve as a building block to further research and more robust evidence on the subject.
Congratulations therefore to the authors. I prompt them to continue to investigate the subject with more patients and more robust methodology and wish all well in health and in their endeavors.
Kind regards,
Dr. Emmanouil I. Kapetanakis
Thoracic Surgeon - Covid-19 ICU Consultant